# Preparation of Imitation Basalt Compound Based on Thermodynamic Calculation

**DOI:** 10.3390/ma12203458

**Published:** 2019-10-22

**Authors:** Shaohan Wang, Xian Luo, Huixin Jin

**Affiliations:** School of Materials and Metallurgy, Guizhou University, Guiyang 550025, China; gzu_wangshaohan@sina.com (S.W.); lxgakki@hotmail.com (X.L.)

**Keywords:** high-temperature melting performance, crystallization ability, imitation basalt compound, thermodynamic calculation

## Abstract

In this paper, imitation basalt compounds using red mud, fly ash or coal gangue as raw materials were designed and prepared with the help of thermodynamic calculations. Thermodynamic calculations were used to obtain the suitable chemical composition. Then, the imitation compounds were prepared and their phase/compositions were analyzed. Finally, their high-temperature melting performance and crystallization ability were evaluated. The results show that the characteristic temperature and crystallization ability of the imitation basalt compounds were similar to those of basalt. Moreover, the viscosity of red mud imitation basalt compound approached the viscosity of basalt with the increase in temperature. This work suggests that red mud, fly ash and coal gangue can be mixed with quartz and other source materials to produce imitation basalt fiber. Therefore, thermodynamic calculation is an effective method to design and prepare high-performance imitation basalt compounds.

## 1. Introduction

Continuous basalt fiber is formed by drawing the basalt melt (at a melting temperature of 1450–1500 °C) with a wire drawing machine. The production process of basalt fiber determines that the production process produces less waste and less environmental pollution. The product is directly degradable in the environment without any hazard and is therefore an environmentally friendly material. Due to its excellent mechanical properties, resistance to chemical erosion, resistance to high temperature and environmental resistance [1], continuous basalt fiber and its composite materials have been extensively used in high-tech fields such as in military materials, transportation, chemical industry, and construction [2]. However, domestic basalt production is in short supply according to available data [3].

Basalt is a natural volcanic rock mainly consisting of SiO_2_, Al_2_O_3_, CaO, Fe_2_O_3_ and MgO; its detailed composition is shown in Table 1. Variations in the chemical composition of basalt lead to an instability of basalt fiber performance, which is not conducive to the industrial production of high-performance basalt fiber. Moreover, the content of iron oxide in basalt ore is generally high, which is unfavorable for the melting of basalt ore and also easily causes the oxidation of the platinum–bismuth alloy leakage plate. As a result, the industrial production of basalt fiber cannot be achieved in a stable and consistent manner [4]. Therefore, it is necessary to prepare high-performance imitation basalt compounds which are suitable for the production of basalt fiber to help the development of the basalt fiber industry.

Red mud is a toxic waste that is discharged from the alumina production process. It cannot be disposed easily due to its alkali content which can contaminate the environment. Its main components are Al_2_O_3_, SiO_2_, CaO and Fe_2_O_3_. During the past century of alumina production, about 2.7 billion tons of red mud was produced [8]. However, the comprehensive utilization rate of red mud was only about 15% [9], which led to a large accumulation of red mud. At present, the cumulative quantity of red mud in China has reached 350 million tons.

Coal gangue is a solid waste generated during coal mining and coal washing, while fly ash is a solid waste generated during thermal power generation. The main components of both coal gangue and fly ash are Al_2_O_3_ and SiO_2_. At present, the cumulative amounts of fly ash and coal gangue in China have reached 1.1 billion tons and 7 billion tons, respectively [10,11]. Such large amounts of waste residues have created tremendous pressure on the environment and have also affected the development of these industries. In recent years, with the improvement of people’s living standards, the state has paid increasing attention to the protection of the natural ecological environment. Thus, the social problems caused by industrial waste such as red mud, fly ash and coal gangue are becoming increasingly acute. In order to alleviate the pressure of red mud, fly ash and coal gangue on the environment and improve their utilization, domestic and foreign researchers have conducted a large number of studies.

For example, Luan, Li et al. [12] used fly ash and CaO as source materials to prepare high-density glass–ceramics with a compressive strength of 328.92 MPa. Liu, Qu et al. [13] prepared aluminosilicate glass with Bayer red mud as the main source material and found that the presence of TiO_2_ and Fe_2_O_3_ in Bayer red mud enhanced the chemical resistance of the glass. Furthermore, Li, Luo et al. [14] prepared foam ceramic glass using coal gangue, quartz, and a sintering agent and found that foamed ceramic glass containing 40% coal gangue exhibited the best performance. In a related study, Chang et al. used red mud to produce imitation basalt fiber and investigated the effects of crystallization and iron oxide on the performance of the material [15,16]. They found that the interaction between iron oxide and titanium dioxide caused the phenomenon of crystallization.

Numerous studies have shown that red mud, fly ash, and coal gangue can be used to produce aluminosilicate glass, which is significant because basalt fiber is a type of aluminosilicate glass [17,18,19,20,21]. Furthermore, Chang et al. have explored the use of red mud to produce imitation basalt fiber [15,16]. However, the use of fly ash and coal gangue for the production of imitation basalt fiber has rarely been reported.

Considering the above background, this paper attempts to make full use of red mud, fly ash, and coal gangue to develop high-performance imitation basalt compounds. For this purpose, thermodynamic calculations were employed to obtain the chemical compositions of imitation basalt compounds with excellent high-temperature melting performance and low crystallization ability. Then, red mud, fly ash, and coal gangue were mixed with quartz, Al_2_O_3_, and other raw materials to prepare the imitation basalt compounds. Finally, experiments were conducted to investigate the differences in the high-temperature melting performance and crystallization ability of basalt and imitation basalt compounds. This research extends the application of red mud to produce basalt fiber and also provides guidance for the production of imitation basalt fiber from fly ash and coal gangue.

## 2. Thermodynamic Calculations

### 2.1. Calculation of Ternary Phase Diagram

The main chemical components of basalt are SiO_2_, Al_2_O_3_, CaO, Fe_2_O_3_ and MgO, with small amounts of K_2_O, Na_2_O and TiO_2_. However, Fe_2_O_3_ and TiO_2_ play a key role in the crystallization process of basalt, which strongly affects the performance of continuous basalt fiber [16]. Therefore, the Phase Diagram module of FactSage7.2 software (Thermfact/CRCT, Montreal, QC, Canada and GTT-Technologies, Achen, Germany) was used to obtain the ternary phase diagram of SiO_2_-Al_2_O_3_-CaO-Fe_2_O_3_-MgO-TiO_2_-O_2_, as shown in Figure 1. Here, the content of MgO, Fe_2_O_3_, and TiO_2_ was the same as that in the basalt sample. Since the points near the phase boundary are not easily crystallized during the cooling process, three points a (0.679 0.106 0.215), b (0.681 0.121 0.198), and c (0.698 0.085 0.217) near the phase boundary in the ternary phase diagram were selected according to the chemical composition of basalt. The sum of the mass ratios of SiO_2_, Al_2_O_3_, and CaO in the imitation basalt compounds is the same as that of the basalt sample. According to the coordinates of the three points a, b, and c in the ternary diagram of SiO_2_-Al_2_O_3_-CaO-Fe_2_O_3_-MgO-TiO_2_-O_2_, the chemical compositions of the imitation basalt compounds were calculated. Table 2 shows the calculated chemical compositions of the obtained imitation basalt compounds, with the same content of K_2_O and Na_2_O as that in the basalt sample.

### 2.2. Calculation of High-Temperature Viscosity

The high-temperature viscosities of basalt and three imitation basalt compounds were calculated by FactSage software. First, the Equilb module of FactSage software was adopted in order to obtain the uniform chemical compositions of the liquid phase in the system under thermodynamic equilibrium conditions. Then, the viscosity module of the FactSage software was applied to calculate the liquid phase chemical compositions obtained from the Equilb module to determine the liquid phase viscosity under the corresponding conditions. If a solid phase existed in the system, the Einstein–Rosce equation was adopted in order to calculate the liquid phase viscosity obtained from the viscosity module, and the solid phase content obtained from the Equilb module was used to obtain the viscosity of the solid–liquid mixture. The calculated viscosities were plotted as a function of temperature, as shown in Figure 2.

The high-temperature viscosity of fly ash imitation basalt compound and coal gangue imitation basalt compound were calculated, and the obtained viscosities were plotted as a function of temperature, as shown in Figure 3.

## 3. Materials and Method

### 3.1. Preparation of Imitation Basalt Compound

The basalt used in the present study was collected from Shandong Juyuan Basalt Fiber Co., Ltd., Zhangqiu, China; the red mud was obtained from Zunyi Alumina Plant, Zunyi, China; the fly ash and coal gangue were from Lingshou County Soil and Mineral Products Processing Factory, Shijiazhuang, China; quartz (AR), alumina (AR), ferric oxide (AR), magnesium oxide (AR), calcium carbonate (AR), sodium carbonate (AR), potassium hydroxide (UR), and titanium dioxide (AR) originated from Baihua Mall (Shanghai, China). In order to remove residual moisture, the raw materials were dried at a temperature of 393 K for 24 h. Following the drying processes, red mud, quartz, alumina, ferric oxide, magnesium oxide, calcium carbonate, potassium hydroxide, and sodium carbonate were, respectively, mixed in a weight ratio of 22.1:48.1:8.8:3.1:5.0:10.0:1.2:3.0 to prepare the red mud imitation basalt compound. Fly ash 1, quartz, alumina, ferric oxide, magnesium oxide, calcium carbonate, potassium hydroxide, sodium carbonate, and titanium dioxide were, respectively, mixed in a weight ratio of 33.0:41.6:0.1:6.1:5.1:11.1:1.4:5.6:0.3 to prepare the fly ash 1 imitation basalt compound. Fly ash 2, quartz, ferric oxide, magnesium oxide, calcium carbonate, potassium hydroxide, sodium carbonate, and titanium dioxide were, respectively, mixed in a weight ratio of 35.0:33.6:6.4:5.2:14.3:1.5:5.6:0.3 to prepare the fly ash 2 imitation basalt compound. Coal gangue, quartz, ferric oxide, magnesium oxide, calcium carbonate, potassium hydroxide, sodium carbonate, and titanium dioxide were, respectively, mixed in a weight ratio of 36.2:34.8:6.5:5.2:15.5:1.5:5.6:0.4 to prepare the coal gangue imitation basalt compound. The prepared imitation basalt compounds have the same chemical composition as the imitation basalt compound b in Table 2. The mixture was ground in a ball mill for 1 h to obtain a homogeneous mixture.

### 3.2. Analysis of Chemical Compositions and Components

The chemical compositions of basalt, red mud, fly ash 1, fly ash 2, coal gangue, quartz, alumina, and the resulting imitation basalt compounds were analyzed by a Zetium X-ray fluorescence spectrometer (Holland Panalytical, Almelo, The Netherlands).

The phase components of basalt red mud, fly ash 1, fly ash 2, and coal gangue were analyzed by X’Pert PRO MPD X-ray diffractometer (Holland Panalytical, Almelo, The Netherlands). 

### 3.3. Investigation of High-Temperature Melting Performance

The softening temperature, hemispherical temperature and flow temperature of the samples were tested by a MTLQ-RD-3 melting point meter (Chongqing University of Science and Technology, Chongqing, China), and the morphology changes of the samples were recorded.

The high-temperature viscosity of basalt and red mud imitation basalt compound were measured by RSV-1600 high-temperature viscosity tester (Orton, Los Angeles, CA, USA), and measured viscosity values were plotted as a function of temperature. 

### 3.4. Investigation of Crystallization Ability

To study their crystallization ability, the samples were first heated from room temperature to 1500 °C by a GSL-1700 high-temperature tube furnace (Hefei Kejing Material Technology Co., Ltd., Hefei, China); then, the temperature was maintained at 1500 °C for 1 h and then cooled to 700 °C at a rate of 5 °C/min. Finally, the samples were cooled to room temperature in the furnace. The crystallization degree of samples was analyzed by X’Pert PRO MPD X-ray diffractometer. The microstructures of the samples after high-temperature treatment were characterized by EM-30 Pluse scanning electron microscope (COXEM, Daejeon, Korea).

## 4. Results and Discussion

Figure 2 shows that the viscosity values of imitation basalt compound b and basalt were relatively close. Therefore, the imitation basalt compound b was prepared.

Figure 4 shows that red mud was mainly composed of serpentine, hematite, gibbsite, hydrogrossular, cancrinite, and biotite. Fly ash 1 was mainly composed of calcite, mullite, and gypsum, while fly ash 2 was mainly composed of mullite and quartz. Coal gangue mainly consisted of muscovite, kaolinite, and quartz. Basalt was mainly composed of diopside, anorthoclase, amphibole, and albite.

Table 3 shows the chemical composition of some raw materials.

Table 4 shows the chemical composition of the imitation basallt compounds.

Figure 5 shows the high-temperature microscopic images and characteristic temperatures of the samples. Characteristic temperatures of fly ash, red mud and coal gangue were much higher than that of basalt, which may be related to their phase components and chemical compositions. Compared with fly ash, red mud, and coal gangue, the characteristic temperatures of the imitation basalt compounds were significantly lower. The characteristic temperature of imitation basalt compounds approached that of basalt, which can be explained by their similar chemical compositions. This indicates that, when the chemical compositions of imitation basalt compounds were similar, the influence of phase components on their characteristic temperature was very limited. Therefore, the characteristic temperatures of imitation basalt compounds prepared by red mud, fly ash, and coal gangue were strongly dependent on their chemical compositions.

Figure 6 indicates that the measured viscosity of red mud imitation basalt compound gradually approaches that of basalt with the increase in temperature. When the temperature range was 1450–1460 °C, the difference in viscosity between the red mud imitation basalt compound and basalt was less than 9 poise. Comparing the calculated viscosities of the imitation basalt compounds in Figure 3 with the measured viscosities of basalt in Figure 6, it can be seen that the calculated viscosities of the imitation basalt compounds decreased rapidly with the increase in temperature, which was consistent with the measured viscosities of the red mud imitation basalt compound. Thus, it can be deduced that the high-temperature melting performance of the imitation basalt compound was strongly dependent on its chemical composition and that imitation basalt compound with excellent high-temperature melting performance can be obtained by thermodynamic calculations.

X-ray diffraction patterns of the samples are shown in Figure 7, which shows that the crystallization behavior of samples cooled from 1500 °C to room temperature. It can be seen that kamaishilite, pyroxene, and merwinite crystals were precipitated out of the red mud melt. However, fly ash and coal gangue were in a solid state at 1500 °C. Therefore, mullite and quartz in the fly ash, anorthite, and alumina in the coal gangue may not be produced by crystallization. It was observed that there was no crystallization during the cooling of basalt and imitation basalt compounds from 1500 °C to room temperature, indicating that both the basalt complex and the basalt have a lower crystallization ability. However, the area of the basalt diffraction peak is smaller than that of the basalt complex, which indicates that the basalt has a lower crystallinity. Similarly, the crystallinity of the red mud-like basalt compound is lower than that of other imitation basalt compound, and the fly ash-like basalt compound has similar crystallinity to the gangue-like basalt compound, which indicates that the crystallization ability of the imitation basalt compound was strongly dependent on its chemical composition. Thus, imitation basalt compounds with low crystallization ability can be obtained by thermodynamic calculations.

Figure 8 indicates that the basalt compound has a uniform matrix composition after high-temperature treatment, and the results of the X-ray diffraction analysis indicate that the basalt compound is amorphous. Therefore, the imitation basalt compound after high-temperature treatment is a uniform amorphous matrix. This indicates that the prepared basalt compound is suitable for the production of basalt fiber.

## 5. Conclusions

(1) The as-prepared imitation basalt compounds exhibited excellent high-temperature melting performance and low crystallization ability, indicating that red mud, fly ash, and coal gangue with different chemical compositions and phase components can be used to produce imitation basalt fiber.

(2) Using thermodynamic calculations to obtain the chemical composition of imitation basalt compounds with desired properties is an effective method for the preparation of high-performance imitation basalt compounds.

(3) The high-temperature melting performance and crystallization ability of the imitation basalt compounds are strongly dependent on their chemical compositions.

## Figures and Tables

**Figure 1 materials-12-03458-f001:**
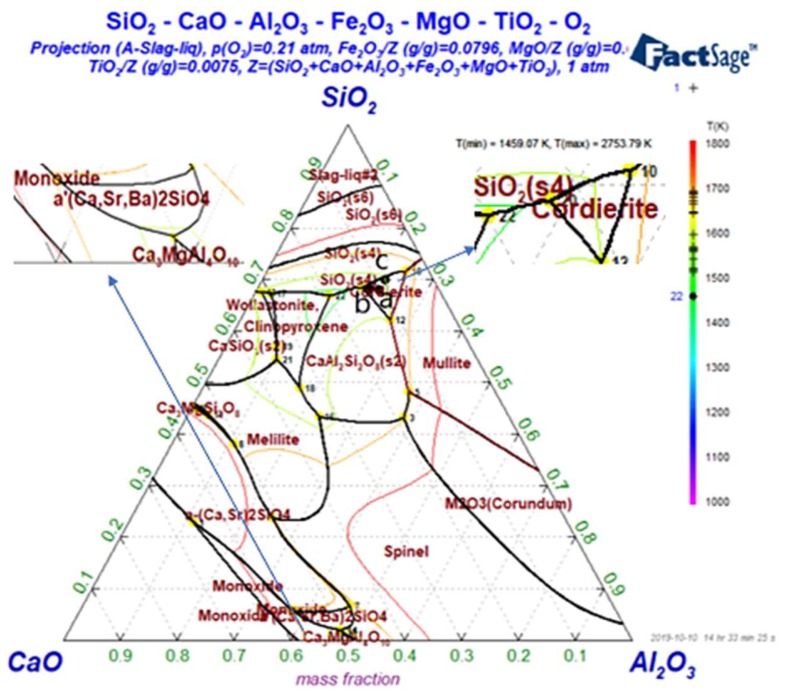
Ternary phase diagram of SiO_2_-Al_2_O_3_-CaO-Fe_2_O_3_-MgO-TiO_2_-O_2_.

**Figure 2 materials-12-03458-f002:**
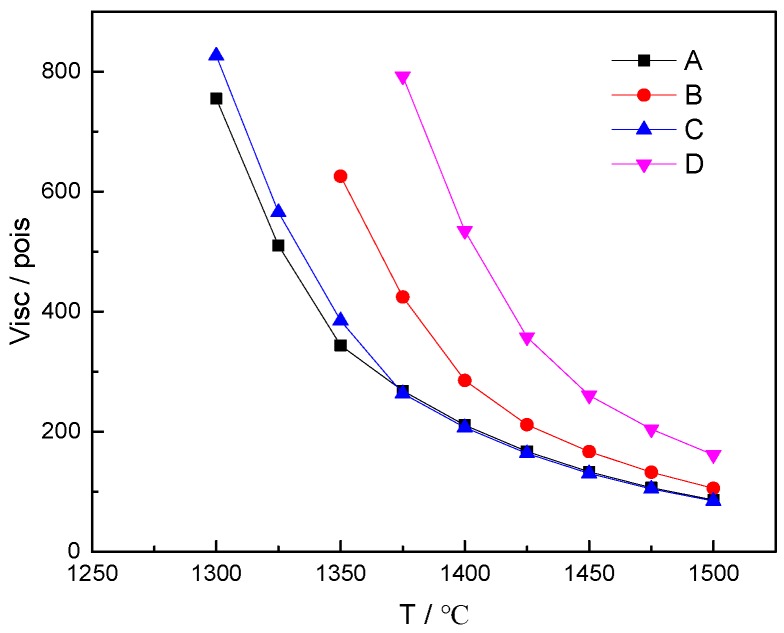
The calculated viscosities of basalt and imitation basalt compounds as a function of temperature: (**A**) basalt; (**B**) imitation basalt compound a; (**C**) imitation basalt compound b; (**D**) imitation basalt compound c.

**Figure 3 materials-12-03458-f003:**
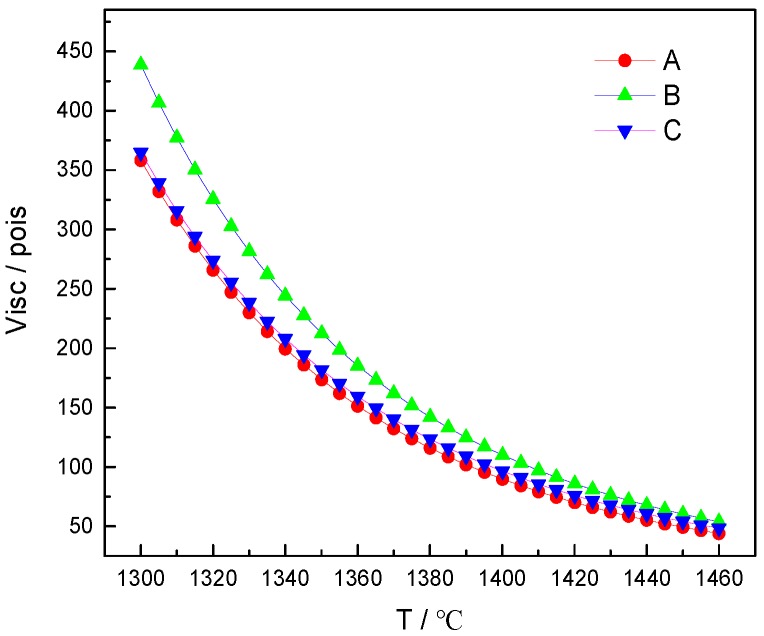
The calculated viscosities of imitation basalt compounds plotted as a function of temperature: (**A**) fly ash 1 imitation basalt compound; (**B**) fly ash 2 imitation basalt compound; (**C**) coal gangue imitation basalt compound.

**Figure 4 materials-12-03458-f004:**
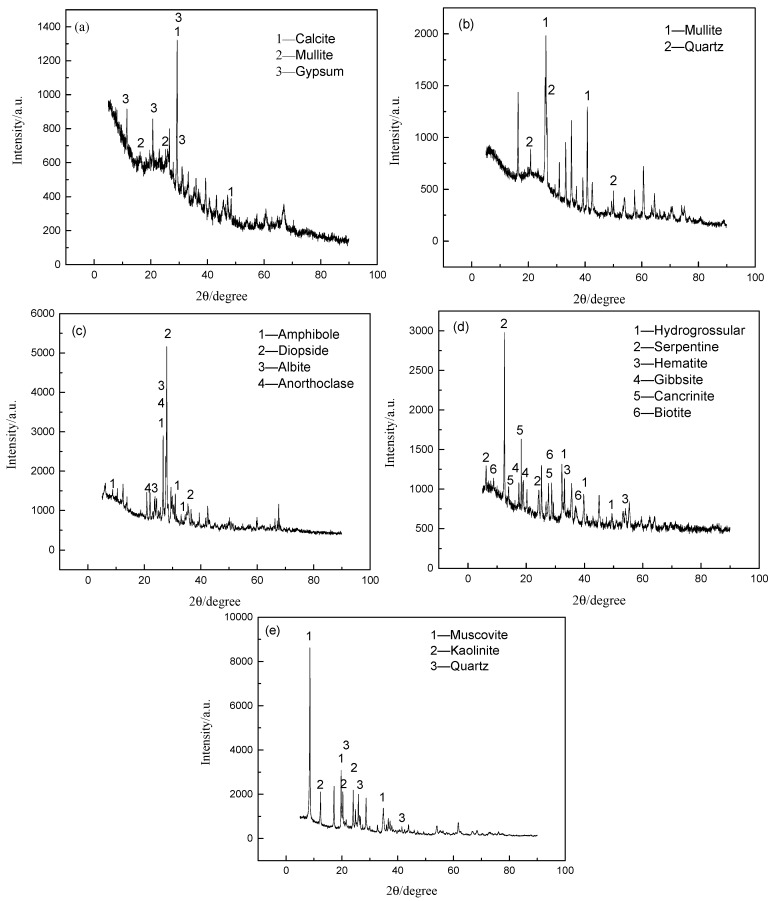
X-ray diffraction patterns of minerals: (**a**) fly ash 1; (**b**) fly ash 2; (**c**) basalt; (**d**) red mud; (**e**) coal gangue.

**Figure 5 materials-12-03458-f005:**
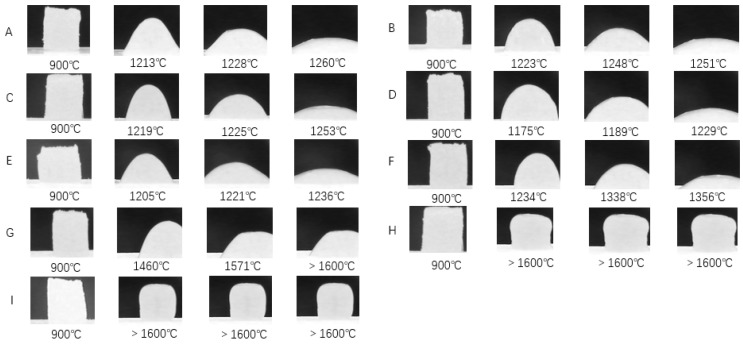
High-temperature microscopic images and characteristic temperatures of the samples: (**A**) fly ash 1 imitation basalt compound; (**B**) coal gangue imitation basalt compound; (**C**) fly ash 2 imitation basalt compound; (**D**) basalt; (**E**) red mud imitation basalt compound; (**F**) red mud; (**G**) fly ash 1; (**H**) coal gangue; (**I**) fly ash 2.

**Figure 6 materials-12-03458-f006:**
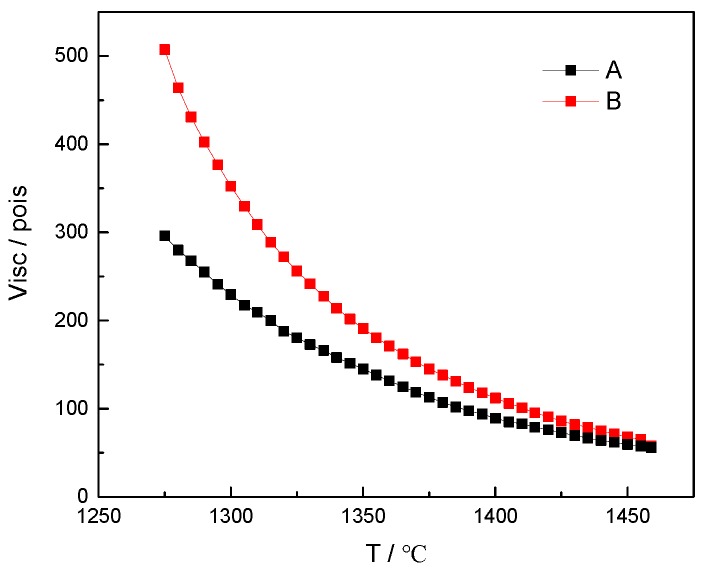
The measured viscosities of basalt and imitation basalt compounds as a function of temperature: (**A**) basalt; (**B**) red mud imitation basalt compound.

**Figure 7 materials-12-03458-f007:**
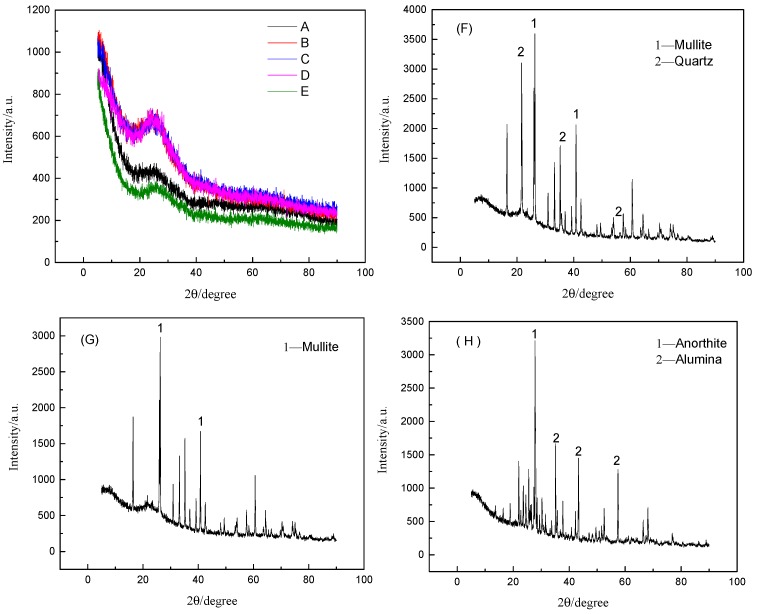
X-ray diffraction patterns of imitation basalt compounds and raw materials after treatment at 1500 °C: (**A**) red mud imitation basalt compound; (**B**) fly ash 1 imitation basalt compound; (**C**) fly ash 2 imitation basalt compound; (**D**) coal gangue imitation basalt compound; (**E**) basalt; (**F**) coal gangue; (**G**) fly ash 2; (**H**) fly ash 1; (**I**) red mud.

**Figure 8 materials-12-03458-f008:**
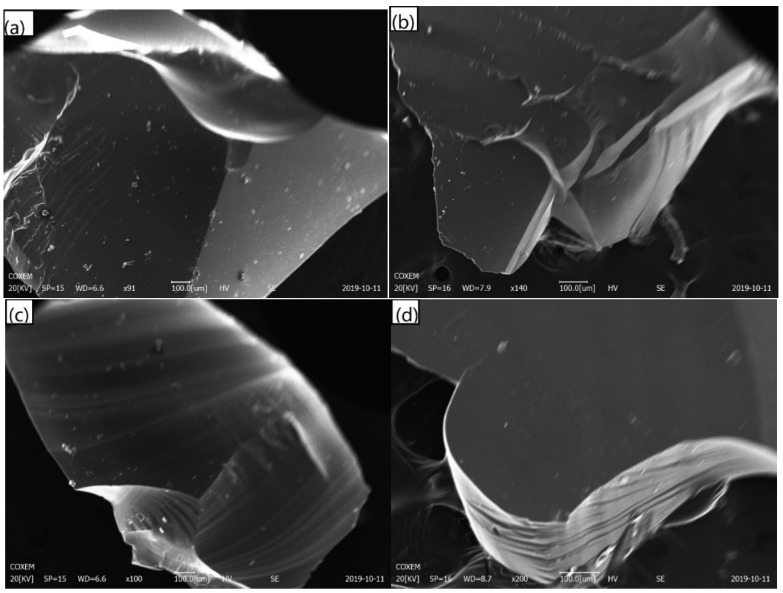
Scanning electron microscope (SEM) image of imitation basalt compound: (**a**) red mud imitation basalt compound; (**b**) fly ash 1 imitation basalt compound; (**c**) fly ash 2 imitation basalt compound; (**d**) coal gangue imitation basalt compound (after high-temperature treatment).

**Table 1 materials-12-03458-t001:** The chemical compositions of basalt (wt.%) [5,6,7].

Component	SiO_2_	Al_2_O_3_	CaO	MgO	K_2_O + Na_2_O	FeO + Fe_2_O_3_	TiO_2_	Others
Maximum	69.0	19.0	13.0	13.0	10.0	16.0	4.0	3.5
Minimum	41.0	10.0	6.0	3.0	2.0	5.0	0.8	2.0

**Table 2 materials-12-03458-t002:** Calculated chemical compositions of imitation basalt compounds (wt.%).

Component	SiO_2_	Al_2_O_3_	CaO	MgO	K_2_O	Na_2_O	Fe_2_O_3_	TiO_2_
Imitation basalt compound a	49.91	15.82	7.74	5.27	1.95	3.33	7.96	0.75
Imitation basalt compound b	50.06	14.53	8.88	5.27	1.95	3.33	7.96	0.75
Imitation basalt compound c	51.28	15.95	6.24	5.27	1.95	3.33	7.96	0.75

**Table 3 materials-12-03458-t003:** The chemical compositions of samples (wt.%).

Component	SiO_2_	Al_2_O_3_	CaO	MgO	K_2_O	Na_2_O	Fe_2_O_3_	TiO_2_
Basalt	50.78	13.59	9.10	5.27	1.95	3.33	7.96	0.75
Red mud	18.00	21.78	14.91	1.06	1.15	6.91	19.79	3.38
Fly ash 1	30.76	40.98	7.94	0.41	0.42	0.1	1.84	1.37
Fly ash 2	50.83	39.55	2.47	0.33	0.63	0.16	2.93	1.22
Coal gangue	45.98	38.06	0.53	0.16	0.41	0.11	0.47	0.86
Quartz	95.45	1.99	0.07	0.00	1.43	0.00	0.83	0.00
Alumina	0.05	98.75	0.05	0.00	0.00	0.55	0.04	0.00

**Table 4 materials-12-03458-t004:** The chemical compositions of imitation basalt compounds (wt.%).

Component	SiO_2_	Al_2_O_3_	CaO	MgO	K_2_O	Na_2_O	Fe_2_O_3_	TiO_2_
Red mud imitation basalt compound	45.47	13.20	8.93	6.72	2.14	2.67	7.41	0.76
Fly ash 1 imitation basalt compound	45.26	13.00	8.88	7.39	2.00	1.59	5.57	0.83
Fly ash 2 imitation basalt compound	45.34	13.42	8.93	7.97	1.98	2.31	6.65	0.82
Coal gangue imitation basalt compound	45.00	12.23	8.74	6.78	1.98	2.80	6.68	0.71

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
