# Peer review of "Preparation of Imitation Basalt Compound Based on Thermodynamic Calculation"

_materials, 2019, doi:10.3390/ma12203458_

Round 1

Reviewer 1 Report

The manuscript is well written, it is clear and concise. The issues are debate with clarity and simplicity.

Figure 1 can be improve

Author Response

Dear Editors and Reviewer:

Thank you for your letter and for the reviewer’s comments concerning our manuscript entitled “Preparation of Imitation Basalt Compound Based on Thermodynamic Calculation” (RMER-D-17-00196). Those comments are all valuable and very helpful for revising and improving our paper, as well as the important guiding significance to our researches. We have studied comments carefully and have made correction which we hope meet with approval. The main corrections in the paper and the responds to the reviewer’s comments are as flowing:

Response to comment: Figure 1 can be improve

Response: I have made improvements to Figure 1 in line 102 of the article.

We appreciate for Editors/Reviewer’s warm work earnestly, and hope that the correction will meet with approval.

Once again, thank you very much for your comments and suggestions.

Reviewer 2 Report

Comments to the paper „Preparation of Imitation Basalt Compound Based on Thermodynamic Calculation” by Shaohan Wang, Xian Luo and Huixin Jin

Paper resents interesting results of investigation of possibility of application of different wastes in production of basalt fiber mater materials. I suggest to relate your results to petrological data related to natural basalts.

Line 22-28: More detail description of applications of continuous basalt fiber will be interesting for readers (statements: military materials, transportation, chemical industry, and construction are to general). Fibrous materials are often considered to be hazardous. Please give evidence that it is “environmentally friendly fiber”.

Line 29: MgO is a very important component (present in pyroxenes, olivines and other minerals in natural basaltic rocks). MgO is mentioned in Table 1 but its role is so important that it should be listed in line 29.

Line 32: What is “basalt ore”? A type of iron ore?

Line 39: Please inform readers what is the reason of the toxicity of red mud? Please add information about toxicity of basalt imitation.

Line 81: I suggest to add MgO among main components. Alkali elements?

Line 92: Phase diagram: please give more information related to this diagram. Why CaO is present in two corners (as CaO and as a component of the sum of SiO2-Al2O3-CaO-Fe2O3-TiO2-O2)? I suggest to add MgO to the calculation of the stability fields in the diagram. The “mineralogy” of basalt imitation is far from “basaltic”

Line 95: Table 2 and Fig. 1. Position of three points a, b and c are different. Please explain why contents of MgO, K2O, Na2O, Fe2O3, TiO2 are the same. Fe2O3 and TiO2 are the components of the sum in the upper corner!

Line 106: Fig. 2 or Fig. 3? Perhaps figures 2 cited in wrong order (Fig. 3 in line 106; Fig. 2 in line 125.

Line 115: In the figure caption there is “fly ash 1 imitation basalt”; in line 135: “red mud imitation basalt”. I suggest to explain clearly the meaning of these terms.

Line 149: Mineral composition of “fly ash 1” – coesite is a high pressure component (above 20 kb) more typical of ultrahigh pressure eclogites than basalts. Chamosite is not a typical component of basalt (Fig. 4c). Is the identification proper? Content of K2O in basalt is low. The presence of microcline needs verification. How it was possible to identify ammonium mica in the coal gangue?

Line 170: Please explain the meaning of diagrams on the Figure 7. Coal gangue composed of mullite and quartz is rather atypical. Similar comment concerns to fly ash composed exclusively of mullite or anorthite and alumina. Careful verification of identification of minerals is suggested. It is possible that more clear description of results is needed. Please explain differences in X-ray diffraction patterns of non-crystalline materials.

Author Response

Dear Editors and Reviewer:

Thank you for your letter and for the reviewer’s comments concerning our manuscript entitled “Preparation of Imitation Basalt Compound Based on Thermodynamic Calculation” (RMER-D-17-00196). Those comments are all valuable and very helpful for revising and improving our paper, as well as the important guiding significance to our researches. We have studied comments carefully and have made correction which we hope meet with approval. The main corrections in the paper and the responds to the reviewer’s comments are as flowing:

Response to comment: Line 22-28: More detail description of applications of continuous basalt fiber will be interesting for readers (statements: military materials, transportation, chemical industry, and construction are to general). Fibrous materials are often considered to be hazardous. Please give evidence that it is “environmentally friendly fiber”.

Response: The production process of continuous basalt fiber determines that the waste generated is small, the environmental pollution is small, and the product can be directly degraded in the environment after being discarded, so it is an environmentally friendly fiber. I have supplemented the environmental friendliness of basalt fiber in lines 22-25 of the article.

Response to comment: Line 29: MgO is a very important component (present in pyroxenes, olivines and other minerals in natural basaltic rocks). MgO is mentioned in Table 1 but its role is so important that it should be listed in line 29.

Response: I have added MgO in line 29 of the article.

Response to comment: Line 32: What is “basalt ore”? A type of iron ore?

Response: Basalt ore refers to basalt that can be used to produce basalt fiber

Response to comment: Line 39: Please inform readers what is the reason of the toxicity of red mud? Please add information about toxicity of basalt imitation.

Response: I have already supplemented the reason why red mud is harmful in the 42 line of the article, but unfortunately I have not found any reports related to the toxicity of basalt imitations.

Response to comment: Line 81: I suggest to add MgO among main components. Alkali elements?

Response: I have added MgO in line 84 of the article.

Response to comment: Line 92: Phase diagram: please give more information related to this diagram. Why CaO is present in two corners (as CaO and as a component of the sum of SiO2-Al2O3-CaO-Fe2O3-TiO2-O2)? I suggest to add MgO to the calculation of the stability fields in the diagram. The “mineralogy” of basalt imitation is far from “basaltic”

Response: The format of references cited in the text has been unified.

Response to comment: Line 95: Table 2 and Fig. 1. Position of three points a, b and c are different. Please explain why contents of MgO, K2O, Na2O, Fe2O3, TiO2 are the same. Fe2O3 and TiO2 are the components of the sum in the upper corner!

Response: When calculating the ternary phase diagram of SiO2-Al2O3-CaO-Fe2O3-TiO2-O2, the content of Fe2O3 and TiO2 in the system is not a variable. Therefore, the three points of a, b, and c in the phase diagram are different but the contents of Fe2O3 and TiO2 are the same. Thermodynamic calculations do not involve three components of MgO, K2O, and Na2O. We directly make the composition of Mg, K2O, and Na2O at a, b, and c the same as those of basalt samples. Therefore, the three positions of a, b, and c in the phase diagram are different but the contents of MgO, K2O, Na2O, Fe2O3 and TiO2 are the same. Fe2O3 and TiO2 are fixed components of the ternary phase diagram

Response to comment: Line 106: Fig. 2 or Fig. 3? Perhaps figures 2 cited in wrong order (Fig. 3 in line 106; Fig. 2 in line 125.

Response: Yes, I have made changes in article 113.

Response to comment: Line 115: In the figure caption there is “fly ash 1 imitation basalt”; in line 135: “red mud imitation basalt”. I suggest to explain clearly the meaning of these terms.

Response: Red mud imitation basalt compounding refers to mixing red mud with other raw materials to prepare a mixture similar to basalt composition. And fly ash 1 imitation basalt compounding refers to mixing fly ash 1 with other raw materials to prepare a mixture similar to basalt composition. I have explained the preparation process of different imitation basalt compound in the 131-141 line of the article.

Response to comment: Line 149: Mineral composition of “fly ash 1” – coesite is a high pressure component (above 20 kb) more typical of ultrahigh pressure eclogites than basalts. Chamosite is not a typical component of basalt (Fig. 4c). Is the identification proper? Content of K2O in basalt is low. The presence of microcline needs verification. How it was possible to identify ammonium mica in the coal gangue?

Response: The phase of fly ash 1, basalt, and coal gangue has been reanalyzed and the previously unsuitable analysis results have been modified in lines 158-169.

Response to comment: Line 170: Please explain the meaning of diagrams on the Figure 7. Coal gangue composed of mullite and quartz is rather atypical. Similar comment concerns to fly ash composed exclusively of mullite or anorthite and alumina. Careful verification of identification of minerals is suggested. It is possible that more clear description of results is needed. Please explain differences in X-ray diffraction patterns of non-crystalline materials.

Response: The xrd diffraction pattern in Fig. 7 is the analysis result of the sample after high temperature calcination, and the phase of the coal gangue may change during the calcination process. In the analysis results, the diffraction peak of coal gangue and the diffraction pattern of mullite and quartz are very stable, so the analysis result of coal gangue has not been changed. Comparative analysis of different amorphous diffraction peaks is supplemented by lines 213-217.

We appreciate for Editors/Reviewer’s warm work earnestly, and hope that the correction will meet with approval.

Once again, thank you very much for your comments and suggestions.

Reviewer 3 Report

The research Wang et al «Preparation of Imitation Basalt Compound Based on 2 Thermodynamic Calculation» is devoted to the task − design and prepare high-performance imitation basalt compounds using red mud, fly ash or coal gangue as raw materials. The authors reveal common patterns of Imitation Basalt Compound structural characteristics changes depending on chemical composition and proportion of initial components. As a whole, the paper contains interesting results that can contribute to the development of industrial technologies of basalt fiber obtaining. Nevertheless, the manuscript demands further improvement to ensure the possibility of its publication.

1) Some figured are of low quality. It causes uneasy analysis of the given results. It refers to the Imitation Basalt Compound and the calculated and measured viscosities of basalt and imitation basalt compounds (Figure 3, Figure 6). The references to picture numbers in the text should be checked (in particular, there probably should be the reference to Fig. 2 inline 106).

2) In unit «3.1. Analysis of chemical compositions and components» there should be a refinement: under which conditions was the sample survey carried out.

3) I recommend that the chemical composition of basalt, red mud, fly ash, coal gangue, quartz and alumina used should be represented in the separate unit «Materials». The same refers to phase composition given in Fig. 4. 

4) The authors should provide a more detailed analysis of the observed regularities of crystallization behavior of the samples imitation basalt compounds in the unit «4. Results and discussion». It’s a really essential part of the work. However, the present manuscript contains a very shallow analysis.

Author Response

Dear Editors and Reviewer:

Thank you for your letter and for the reviewer’s comments concerning our manuscript entitled “Preparation of Imitation Basalt Compound Based on Thermodynamic Calculation” (RMER-D-17-00196). Those comments are all valuable and very helpful for revising and improving our paper, as well as the important guiding significance to our researches. We have studied comments carefully and have made correction which we hope meet with approval. The main corrections in the paper and the responds to the reviewer’s comments are as flowing:

Response to comment: 1) Some figured are of low quality. It causes uneasy analysis of the given results. It refers to the Imitation Basalt Compound and the calculated and measured viscosities of basalt and imitation basalt compounds (Figure 3, Figure 6). The references to picture numbers in the text should be checked (in particular, there probably should be the reference to Fig. 2 inline 106).

Response: Changes have been made to Picture 3 and Picture 6 on lines 123 and 198.

Response to comment: 2) In unit «3.1. Analysis of chemical compositions and components» there should be a refinement: under which conditions was the sample survey carried out.

Response: The results of the sample survey and experimental methods are fully described in lines 129-147.

Response to comment: 3) I recommend that the chemical composition of basalt, red mud, fly ash, coal gangue, quartz and alumina used should be represented in the separate unit «Materials». The same refers to phase composition given in Fig. 4.

Response: The chemical composition analysis and phase composition at lines 139-154 are already located in the separate unit «Material and method».

Response to comment: 4) The authors should provide a more detailed analysis of the observed regularities of crystallization behavior of the samples imitation basalt compounds in the unit «4. Results and discussion». It’s a really essential part of the work. However, the present manuscript contains a very shallow analysis.

Response: The microstructure of the sample was examined by SEM in lines 233-234, and the microstructure of the sample was described and analyzed in lines 238-242.

We appreciate for Editors/Reviewer’s warm work earnestly, and hope that the correction will meet with approval.

Once again, thank you very much for your comments and suggestions.

Round 2

Reviewer 3 Report

Accept in present form